# A Subset of Non-Small Cell Lung Cancer Patients Treated with Pemetrexed Show ^18^F-Fluorothymidine “Flare” on Positron Emission Tomography

**DOI:** 10.3390/cancers15143718

**Published:** 2023-07-22

**Authors:** Preetha Aravind, Sanjay Popat, Tara D. Barwick, Neil Soneji, Mark Lythgoe, Katherina B. Sreter, Jingky P. Lozano-Kuehne, Mattias Bergqvist, Neva Patel, Eric O. Aboagye, Laura M. Kenny

**Affiliations:** 1Department of Surgery and Cancer, Faculty of Medicine, Hammersmith Hospital Campus, Imperial College London, Du Cane Road, London W12 0NN, UK; p.aravind@imperial.ac.uk (P.A.); tara.barwick@nhs.net (T.D.B.); neil.soneji@nhs.net (N.S.); mark.lythgoe@nhs.net (M.L.); j.lozano-kuehne@imperial.ac.uk (J.P.L.-K.); neva.patel1@nhs.net (N.P.); 2Lung Unit, The Royal Marsden NHS Foundation Trust, Fulham Road, London SW3 6JJ, UK; sanjay.popat@rmh.nhs.uk (S.P.); katherina.sreter@kbcsm.hr (K.B.S.); 3Department of Imaging, Charing Cross Hospital, Imperial College Healthcare NHS Trust, Fulham Palace Road, London W6 8RF, UK; 4Biovica International, Uppsala Science Park, 75237 Uppsala, Sweden; mattias.bergqvist@biovica.com; 5Department of Medical Oncology, Charing Cross Hospital, Imperial College Healthcare NHS Trust, Fulham Palace Road, London W6 8RF, UK

**Keywords:** NSCLC, ^18^F-FLT, PET, pemetrexed, thymidine kinase

## Abstract

**Simple Summary:**

Thymidylate synthase (TS) inhibitors have remained among the most effective chemotherapies used in the treatment of various cancer types. Imaging of tumour proliferative activity while on antifolates has been studied with 3′-deoxy-3′-[^18^F]fluorothymidine positron emission tomography (^18^F-FLT PET) imaging. The aim of this study was to use ^18^F-FLT PET/CT imaging to understand the basis of early drug action with the antifolate drug pemetrexed on TS inhibition. While every patient showed a global change in the plasma marker of TS inhibition after drug administration, tumour TS inhibition was selective and patients who showed a tumour change in the imaging biomarker had a greater therapy response and longer overall survival following a combination treatment including pemetrexed. The study findings implicate the potential use of ^18^F-FLT PET/CT to understand the basis of drug action in other studies involving TS inhibitors.

**Abstract:**

Thymidylate synthase (TS) remains a major target for cancer therapy. TS inhibition elicits increases in DNA salvage pathway activity, detected as a transient compensatory “flare” in 3′-deoxy-3′-[^18^F]fluorothymidine positron emission tomography (^18^F-FLT PET). We determined the magnitude of the ^18^F-FLT flare in non-small cell lung cancer (NSCLC) patients treated with the antifolate pemetrexed in relation to clinical outcome. Method: Twenty-one patients with advanced/metastatic non-small cell lung cancer (NSCLC) scheduled to receive palliative pemetrexed ± platinum-based chemotherapy underwent ^18^F-FLT PET at baseline and 4 h after initiating single-agent pemetrexed. Plasma deoxyuridine (dUrd) levels and thymidine kinase 1 (TK1) activity were measured before each scan. Patients were then treated with the combination therapy. The ^18^F-FLT PET variables were compared to RECIST 1.1 and overall survival (OS). Results: Nineteen patients had evaluable PET scans at both time points. A total of 32% (6/19) of patients showed ^18^F-FLT flares (>20% change in SUVmax-wsum). At the lesion level, only one patient had an FLT flare in all the lesions above (test–retest borders). The remaining had varied uptake. An ^18^F-FLT flare occurred in all lesions in 1 patient, while another patient had an ^18^F-FLT reduction in all lesions; 17 patients showed varied lesion uptake. All patients showed global TS inhibition reflected in plasma dUrd levels (*p* < 0.001) and ^18^F-FLT flares of TS-responsive normal tissues including small bowel and bone marrow (*p* = 0.004 each). Notably, 83% (5/6) of patients who exhibited ^18^F-FLT flares were also RECIST responders with a median OS of 31 m, unlike patients who did not exhibit ^18^F-FLT flares (15 m). Baseline plasma TK1 was prognostic of survival but its activity remained unchanged following treatment. Conclusions: The better radiological response and longer survival observed in patients with an ^18^F-FLT flare suggest the efficacy of the tracer as an indicator of the early therapeutic response to pemetrexed in NSCLC.

## 1. Introduction

Lung cancer is the second most diagnosed cancer and the leading cause of cancer death [1]. Non-small cell lung cancer (NSCLC) accounts for 85% of all lung cancers. In the changing landscape of treatments, platinum-based combination therapies with pemetrexed continue to be the standard of care along with immunotherapy [2,3], and patient selection for pemetrexed maintenance in the chemo-immunotherapy era remains an important question. Thymidylate synthase (TS) is a critical enzyme for DNA replication catalysing the de novo synthesis of pyrimidine nucleotides. Since the 1940s, inhibitors of thymidine biosynthesis and TS, including classical and non-classical TS inhibitors, have remained among the most effective chemotherapies used in the treatment of various cancer types [4]. Pemetrexed is a multi-targeted antifolate anti-cancer agent that inhibits TS, dihydrofolate reductase (DHFR), and glycinamide ribonucleotide formyl transferase (GARFT) [5,6]. At least two aspects of TS research seek to use tumour biomarkers to predict response. The first involves determining the pharmacodynamics of new TS inhibitors, including α-folate-targeted TS inhibitors [7], nucleoside transporter-independent TS inhibitors [8], and new screening hits [9]. The other involves future efforts directed at optimising efficacy through understanding and modulating target enzyme expression/activity, including post-translational modification of the enzyme via *O*-GlcNAcylation [10]. Selective imaging of tumour tissue TS inhibition, together with information on enzyme inhibition in diverse TS-responsive healthy tissues, will enable a therapeutic index to be predicted.

The imaging of tumour proliferative activity has been studied with 3′-deoxy-3′-[^18^F]fluorothymidine (^18^F-FLT) positron emission tomography (PET) scans in various tumour types including NSCLC [11]. Beyond proliferation imaging, we previously demonstrated that tumour cells can translocate the equilibrative nucleoside transporter (ENT1) to the cell membrane within minutes to hours in response to TS inhibition, allowing increased uptake of thymidine nucleosides—including ^18^F-FLT—via the salvage pathway for their activation by thymidine kinase 1 (TK1), resulting in a transient “flare” response [12]. Such changes are concomitant with the block in the utilisation of nucleosides following TS inhibition, leading to increases in deoxyuridine (dUrd) [12]. The inhibition of TS is associated with the accumulation of precursors of the de novo thymidine monophosphate synthesis pathway, including dUrd which can diffuse into plasma. This has enabled the use of the change in plasma dUrd concentration as a surrogate biomarker of TS inhibition [13]. The use of nucleosides such as ^18^F-FLT to detect TS inhibition is only possible during early “on-drug” pharmacodynamics assessment [14], as decreases in ^18^F-FLT resulting from decreased proliferation could occur over time. Appropriate timing of an ^18^F-FLT PET after the administration of pemetrexed is a clinical unknown. In translating our understanding from preclinical studies, a pilot study performed in patients with breast cancer by our group using the TS inhibitor capecitabine demonstrated increased uptake in tumour lesions, compared to baseline, at 1 h after initiating treatment, demonstrating feasibility for the approach in humans [15]. Furthermore, ^18^F-FLT flare effects have been reported in mouse models of cancer for the TS inhibitors plevitrexed and BGC945/CT900 at 4 and 24 h, respectively [14]. A clinical study of CT900 in seven patients showed increases in ^18^F-FLT tumour uptake at 16–24 h [7]. Initial studies of pemetrexed at 4 h also showed flare effects; however, the patient-level tumour ^18^F-FLT flare was unrelated to clinical outcome [16].

In view of the potential utility that a clinical TS inhibitor imaging assay will provide the research community, we conducted a detailed lesion assessment of ^18^F-FLT flares to investigate whether the TS inhibitor pharmacodynamic imaging technique is a reliable early predictor of treatment response and survival in NSCLC patients scheduled for treatment with pemetrexed ± platinum, using the timing informed by our preclinical studies [14] and initial clinical studies [16]. This study represents the first analytical approach that shows all lesions within a patient as a weighted sum variable.

## 2. Study Design

### 2.1. Patients

We studied twenty-one patients attending oncology clinics at Imperial College Healthcare NHS Trust and the Royal Marsden Hospital NHS Trust, London, over a period of 3 years (2011–2014). All patients included were aged ≥18 years, of good performance status (ECOG 0–2), and had clinically acceptable blood parameters and tumour molecular analysis results collected from that performed for routine clinical care. All had at least one measurable site of disease ≥2 cm (primary tumour or lymph node) and were scheduled for treatment with pemetrexed ± platinum. Ethical approval was granted for this study by the Hammersmith Hospital Research Ethics Committee (10/H1109/40) and the administration of radioactivity was approved by the Administration of Radioactive Substances Advisory Committee (ARSAC). All patients provided written informed consent in accordance with the Declaration of Helsinki.

Baseline ^18^F-FLT PET scans were obtained within one week before the start of treatment with pemetrexed followed by a similar scan conducted approximately 4 h after the first therapeutic pemetrexed dose (500 mg/m^2^). Combination therapy with either cisplatin or carboplatin was given the following day to avoid interference with ^18^F-FLT uptake. Patients continued to receive the chemotherapy schedule as per the standard of care of 3 weekly cycles.

### 2.2. PET Protocol

The ^18^F-FLT used in this study was manufactured according to standard protocols. All patients were scanned on a Siemens Biograph 64-slice PET/CT scanner. All patients received a single bolus intravenous injection of ^18^F-FLT (mean: 208 (range 150–238) MBq) over 30 s, followed by dynamic single-bed list mode acquisition (thoracic/abdominal to cover the primary tumour, regional lymph nodes, and liver) for 66 min, and a whole-body scan (vertex to thighs) for 30 min commencing at approximately 90 min after radiotracer injection. These were each preceded by a CT scan (50 mA, 110 kVp, 0.8 pitch, 0.6 s/rotation) for both attenuation correction and co-registration with PET images to allow good anatomical visualisation and localisation of ^18^F-FLT activity. Raw PET data were corrected for scatter and attenuation and reconstructed with an iterative algorithm using ordered subset expectation maximisation reconstruction with 8 iterations and 16 subsets. The data were binned into time frames as follows: 1 × 30 (background), 6 × 10, 4 × 20, 4 × 30, 5 × 120, 4 × 180, and 4 × 600 s.

### 2.3. PET Analysis

Volumes of interest (VOIs) were defined on a Hermes workstation (Hermes Diagnostics, Stockholm, Sweden) by a single experienced observer (TDB, 20 years’ experience) around tumour index lesions and normal background organs. VOIs were drawn on sites of identifiable tumour uptake >20 mm size using a semiautomatic threshold technique, with 40% of the SUVmax threshold with manual adjustment if required. SUV40% is a routinely used auto-segmentation adaptive threshold target delineation method in semi-quantitative PET analysis and available on commercial software platforms such as that used here (Hermes Gold3; Hermes Medical Solutions Ltd., London, UK). The method has been shown to be robust in phantom studies [17]. Mean and maximum voxel standardised uptake values (SUVs) were determined at 60 min normalised to body weight (SUV60_mean_ and SUV60_max_) on baseline and post-treatment ^18^F-FLT PET/CT studies. The percentage change in SUV in both SUV_mean_ and SUV_max_ was then calculated for each target lesion visible on baseline imaging as follows:ΔSUV_max_ = 100 × SUV_max_(PET2) – SUV_max_(PET1)/SUV_max_(PET1)

In addition, mean SUVs corrected for bodyweight were calculated for normal background non-tumour reference tissues: 3 cm VOI liver, small bowel and lung; 2 cm VOI mediastinal blood pool, bone marrow.

To obtain a measure of the combined uptake in all lesions within a patient, we calculated the ΔSUV_max_ variable as the weighted sum of the ΔSUV_max_ values in all lesions according to Equation (1).
ΔSUV_max_ wsum = Σi (wi × ΔSUV_max_ i)/Σi wi with i = [1, N]…(1)
with N = number of lesions.

In the equation above, wi is the weight relative to the i-th lesion (with a ΔSUV_max_ = ΔSUV_max_ i). Weights were proportional to the uptake of the lesion (e.g., w = 1 for the hottest lesion). The sum-weighted PET variable was used to depict patient-level disease burden with which to correlate patient-level variables (plasma dUrd levels and plasma TK1 activity).

The patients’ standard imaging contrast-enhanced CT (CECT) scans were evaluated by one experienced observer blinded to the PET data (N.S., 10 years’ experience) to assess response by Response Evaluation Criteria in Solid Tumours version 1.1 RECIST1.1 [18].

The repeatability of quantitative uptake measures with the ^18^F- fluorodeoxyglucose (^18^F-FDG) PET that has been integrated into the response assessment criteria Positron Emission Tomography Response Criteria in Solid Tumours (PERCIST) [19], as well as that of ^18^F-FLT, is deemed to be about 20% [20,21,22]. To assess the true change in tumour uptake at the patient level, we used a cut-off 20%, i.e., patients with an increase >20% were deemed to have an ^18^F-FLT flare [20,21,22].

### 2.4. Analysis of Plasma Thymidine Kinase 1 (Plasma TK1) and Plasma Deoxy Uridine (Plasma dUrd)

Plasma samples for TK1 and dUrd were collected at the baseline scan visit, 4 h after the administration of pemetrexed, and prior to the injection of the ^18^F-FLT radiotracer. Plasma TK1 activity was determined via the DiviTum Tka assay [23], a refined ELISA-based method, at Biovica laboratories (Uppsala, Sweden). Analysis was blinded. The working range of the assay is 100–2000 Du; the coefficient of variation (CV) is <20% at 100 DuL. Durd (ng/mL) was measured using a validated liquid chromatography with tandem mass spectrometry detection (LC-MS/MS) assay [24] under contract by the Kymos group (Barcelona, Spain). The method was validated in the range of 5 to 400 nM for 2′-deoxyuridine. The lower limit of quantification was set at 5 nM.

### 2.5. Statistical Analysis

Statistical analysis was performed using GraphPad Prism version 8.0 (GraphPad Software, San Diego, CA, USA) and SPSS for Mac version 27 (SPSS, Chicago, IL, USA). Radiotracer uptake was determined at the patient level (n = 19) and median percentage differences in SUV_mean_ and SUV_max_ were determined at the lesion level (n = 61). The statistical analysis to determine changes and compare groups consisted of the Wilcoxon signed-rank test, Kruskal–Wallis test, and paired t-test. The association between PET parameters, plasma TK1, and plasma dUrd were determined by calculating the Pearson correlation coefficient (95% confidence interval; two-tailed); *p* ≤ 0.05 was considered significant. The differences in ^18^F-FLT uptake were correlated with clinical outcome measured as tumour response. Response was evaluated according to RECIST 1.1. TTP and OS were calculated at a median of 10 years of follow up. The median value for time to tumour progression (TTP, defined as date of treatment start to date of disease progression) and overall survival (OS, defined as date of diagnosis to date of death) were calculated using the Kaplan–Meier method and *p*-values were derived with the log rank test.

## 3. Results

### 3.1. Demographics

Twenty-one NSCLC patients with Stage III and IV disease were included in the study (Table 1). All patients tolerated the PET scanning protocol, and none reported adverse events due to the study imaging intervention. The median age was 62 years (range 38–78) and nine patients (43%) were male. Twenty patients were chemotherapy naïve and one patient had prior treatment with gefitinib. Three patients had prior radiotherapy targeted on the index lesions which were classified by the RECIST criteria. After study enrolment, nine patients received pemetrexed in combination with carboplatin, twelve with cisplatin combination, and four continued on pemetrexed maintenance monotherapy per routine clinical decision making. Of the 21 patients recruited to the study, 19 had both ^18^F-FLT PET static scans for analysis. Two patients were not evaluable for dynamic scan results due to technical issues with the PET scanner and two patients withdrew consent after the first scan (Figure 1A).

We analysed the ^18^F-FLT uptake in individual tumour lesions (primary and metastatic lymph nodes, n = 61 lesions). Figure 1B,C show an example of ^18^F-FLT PET/CT scans of two patients with increased ^18^F-FLT uptake and another patient with decreased uptake before and after pemetrexed administration.

The SUV differences for individual lesions are shown in Figure 2A,B. We found no differences in the overall change in radiotracer uptake at the patient level. The median percentage differences of SUV_mean_ and SUV_max_ in all tumour lesions combined increased by 0.02% and 8%, respectively.

On the contrary, there was a flare effect at the lesion level. Based on the ∆SUV_max_-wsum, 32% (6/19) of patients showed a significantly increased ^18^F-FLT tumour uptake 4 h after therapy compared with baseline (beyond test–retest borders of 20%) [20,22]. In the remaining thirteen patients, the change in ^18^F-FLT uptake was within the test–retest variability <20%. No additional significant change in SUV_max_ was seen in the dynamic images as estimated at the 60:1 min and 60:5 min time points (Appendix A).

In the lesion-level analysis, 1 patient had ^18^F-FLT uptake increased in all the individual lesions at 4 h after pemetrexed, 1 patient had a decrease in all lesions, and the remaining 17 patients had varied uptake among the individual lesions (Figure 2B).

### 3.2. Analysis by Plasma dUrd

A global change in plasma dUrd levels was detected between the two time points (Figure 2C). The baseline plasma dUrd had a median of 37.7 ± 9.31 ng/mL. At 4 h after administration of pemetrexed, plasma dUrd levels significantly rose in all patients (median of 96.0, *p* < 0.001) (Figure 2C). No significant difference was seen with FLT flare (Figure 2D) and no correlation was found between the ∆plasma dUrd and ∆SUV_max_ (Figure 2F).

### 3.3. Clinical Outcome

The RECIST-defined response comparing the baseline CT to the 9-week follow-up CT demonstrated a partial response in 4 patients, stable disease in 11, and progressive disease in 4. No association was noted between treatment response and baseline SUV_mean_ and SUV_max_. Of the six patients who had an ^18^F-FLT flare above the threshold, five were classified as (RECIST) responders (Figure 3A,B). The ∆SUV_max_ was not related to RECIST response (*p* = 0.230).

Median TTP and OS were 5.1 m (range 1.1–26.0 m) and 17.7 m (range 0.9–94.0) for the entire group of patients (n = 21). The median OS in the patients with ^18^F-FLT flares (n = 6) was 31.0 m, unlike the group without ^18^F-FLT flares (<20%, n = 13) (median OS, 15.0 m) (Figure 3C). The ∆SUV_max_ was not associated with survival (*p* = 0.152) (Appendix A). Interestingly, patients who showed response on RECIST also had a higher plasma dUrd change, indicating effective global TS inhibition (Figure 2E).

### 3.4. Pemetrexed Increased ^18^F-FLT Uptake in TS-Responsive Healthy Tissues

A flare effect was also observed in TS-responsive normal tissue, as shown by the SUV_mean_ and SUV_max_ in Figure 4A. Both bowel tissue and bone marrow have physiologically high rates of cell turnover and are TS responsive [12,14]. The median SUV_max_ and SUV_mean_ for background bowel tissue were 2.7 and 1.9 for baseline and increased to 4.12 and 2.83 at 4 h, which was statistically significant (*p* = 0.004 and 0.004, respectively). The median SUV_max_ and SUV_mean_ for background bone marrow were 9.2 and 6.5 at baseline and increased to 14.5 and 9.16 at 4 h, which was also statistically significant (*p* = 0.004 and 0.005, respectively), in keeping with the previous studies [14,16]. There was decreased ^18^F-FLT uptake in background liver tissue and no change in the ^18^F-FLT uptake in normal lung tissue (Figure 4B). The uptake in normal tissue together with lesion changes paralleled systemic changes in dUrd. Of note, the magnitude of SUV variables in TS-responsive tissues at baseline was found to be higher in the tumour flare group (>20%) compared to those without flares (<20%) (Appendix A). We do not understand the basis for this.

### 3.5. Plasma TK1 Activity Is Unchanged with Pemetrexed Treatment

We considered if the increase in tissue ^18^F-FLT could be explained in part by increases in TK1 activity. To examine the role of TK1 in ^18^F-FLT flares, we measured plasma TK1 activity before and after treatment with pemetrexed. The baseline plasma TK1 activity showed a wide range of values (range 99–2000 DuA). No change in plasma TK1 activity was observed at 4 h after administration of pemetrexed (*p* = 0.6, paired *t*-test) (Figure 5A). No significant correlation was found between ∆plasma TK1 and ∆SUV_max_-wsum (*p* = 0.546) (Figure 5B). Interestingly, however, baseline plasma TK1 activity correlated strongly with overall survival (68 m with plasma TK1 activity <105 DuA vs. 25 m in plasma activity >105 DuA, log rank *p* = 0.02), as seen in Figure 5C.

## 4. Discussion

Pemetrexed is used routinely with platinum-based chemotherapy in non-squamous lung cancer and thereafter as a first-line maintenance therapy. However, there is a dearth of predictive biomarkers that will help determine optimal patient selection for pemetrexed, especially in the era of immunotherapy where hepatic and renal toxicities from the immunotherapy–pemetrexed combination can be problematic and where chemotherapy treatment selection in the PD-L1 ≥ 50% subgroup remains imprecise, along with intrinsic and acquired treatment resistance to pemetrexed and the utility of future TS inhibitors under development. In the future, novel imaging biomarkers could also help decide which patients should be offered alternative therapies early, or which patients benefit little from pemetrexed, where an immune checkpoint inhibitor only based strategy may be preferred.

In this study, we have explored the activity of the thymidine salvage pathway in investigating early lesion pharmacodynamic response reflected in ^18^F-FLT flares with pemetrexed. We show that ^18^F-FLT PET flares occur in a subset of NSCLC patients undergoing treatment with pemetrexed ± platinum and that patients who demonstrate the ^18^F-FLT flare showed survival two times longer than those not showing a flare.

Prior studies by Kenny et al. showed that of the six patients with breast cancer who were treated with the TS inhibitor capecitabine, the ^18^F-FLT uptake kinetics increased at 1 h after treatment [15]. However, the study by Frings et al. with pemetrexed in NSCLC showed no association of increased ^18^F-FLT with outcome when patient-level flare was investigated [16]. Furthermore, they reported the increased uptake at 4 h post pemetrexed in two patients who showed an SUV_max_ increase (31% and 35%, respectively), whereas two other patients showed a 35% decrease in uptake [16]. One possibility for the positive outcome of our study compared to the outcome reported by Frings et al. is the investigation at an individual lesion level since we also did not identify overall patient-level changes. Whether further interventions for lesions that do not flare would lead to improved patient outcomes remains to be seen. Recently, Banerji et al. showed in a clinical study on ovarian cancer that a subset of patients who were treated with BCG945/CT400 (a novel alpha FR targeted TS inhibitor) and underwent pre- and post-treatment ^18^F-FLT PET scans had increases in ^18^F-FLT PET signals in the tumour tissue at 16–24 h post treatment, consistent with TS inhibition in tumour tissue [7]. The flare effect among patients was independent of the global increases in dUrd and flare of TS-responsive healthy tissues. Our data are consistent with that of Frings et al., demonstrating changes in TS-responsive healthy tissues and plasma dUrd in all patients [16].

Preclinical studies have shown that tumour-selective alpha-folate-targeted inhibitors only display flares in tumours, compared to TS inhibitors that target both the reduced folate carrier (RFC) and alpha folate transporters [14]. An increase in uptake is expected in responding patients if clinical response is mainly attributed to TS inhibition. Our study has further demonstrated that TS inhibition on target lesions is not uniform. Only six out of nineteen patients (32%) demonstrated an increased ^18^F-FLT uptake 4 h after administration of pemetrexed beyond test–retest boundaries (20% for SUV_max_) for wsum in each patient. Five of the six (83%) patients with an increase in ∆SUV_max_-wsum were responders measured with RECIST after 9 weeks. Of clinical relevance, patients who showed ^18^F-FLT flares had twice as long median OS (31 months versus 15 months). The data presented support the notion that an FLT flare may have clinical use as a biomarker, due to longer OS in the flare vs. non-flare groups, respectively (OS: 31 vs. 15).

The rationale for using the change in the weighted summed variable (∆SUV_max_-wsum), by combining values from all the lesions in a weighted manner as an analytical method, was to reflect the aggregate flare response. This variable needs to be explored in future studies to evaluate the heterogeneity among multiple lesions.

No significant change in plasma TK1 was seen in the treatment within 4 h of pemetrexed. This is most likely since changes in plasma protein expression are unlikely to occur in a short time interval. Thus, the flare effect is unlikely to be due to changes in TK1 as this was not seen even in easily accessible blood. Our previous study further demonstrated the redistribution of ENT1 to the cell surface for the increased transport of thymidine into the cell to activate the TK1-catalysed salvage pathway [12]; we did not assess ENT1 levels, which are known to be redistributed to the cell surface [25]. Plasma TK1 has previously been shown to be a prognostic marker for survival as it is a reflection of the tumour burden [26,27,28] and was similarly demonstrated in our study independently of its effect on ^18^F-FLT uptake.

We demonstrated early effective TS inhibition with increased ^18^F-FLT uptake in the small bowel within 4 h of pemetrexed administration, in line with preclinical studies by Perumal et al. who also showed differential uptake with the use of selective and non-selective TS inhibitors [12]. The ^18^F-FLT uptake also increased in bone marrow, which is supported by the study from Frings et al., suggesting that bone marrow and the small bowel are possible surrogate tissues to assess sub-clinical toxicity from pemetrexed [16]. The decreased uptake in the liver despite the limited hepatic metabolism of pemetrexed is likely explained by the effect on glucuronidation [29]. A limitation of our study is the small number of patients analysed.

Overall, the magnitude of ^18^F-FLT flares detected among the subset of patients has the potential to be used in the future to develop new TS inhibitors, evaluate the effect of modulators [9,10], and to evaluate further strategies on the personalisation of pemetrexed–immunotherapy use in advanced non-squamous NSCLC.

## 5. Conclusions

Effective TS inhibition was seen at 4 h post pemetrexed in TS-responsive normal tissues and in tumours. The subset of patients who showed a tumour change in the imaging biomarker had better response and longer survival following a combination treatment with pemetrexed. This effect could be used to understand the basis of the drug action of TS inhibitors and modulators.

## Figures and Tables

**Figure 1 cancers-15-03718-f001:**
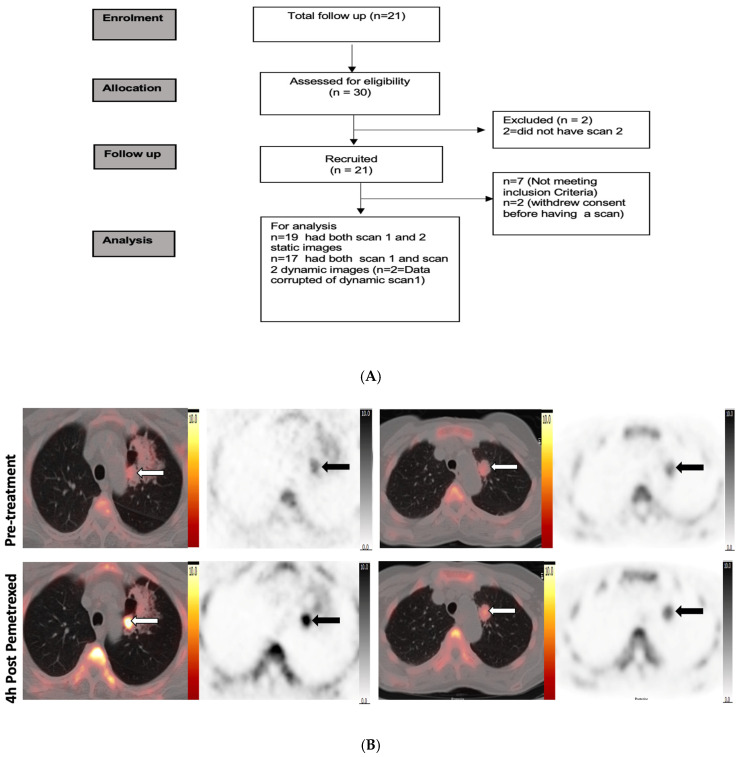
Study design and PET images. (**A**) Consort diagram of the PET study and summary of patients. (**B**) Representative ^18^F-FLT PET/CT images of two responding patients acquired at baseline and 4 h post pemetrexed demonstrating increase in ^18^F-FLT radiotracer uptake in the left upper lobe primary tumour (white arrow on fused images, black arrow in PET image). Increased uptake in the tumour represented by white arrow in fused I mage and black arrow in the PET image. (**C**) Representative images of a non-responding patient’s ^18^F-FLT PET/CT scan acquired at baseline and 4 h post pemetrexed demonstrating decrease in ^18^F-FLT radiotracer uptake in the coeliac node and right lower lobe lung primary tumour (white arrow on fused images, black arrow in PET image); also noted is the increase in uptake in background bone marrow (green arrow).

**Figure 2 cancers-15-03718-f002:**
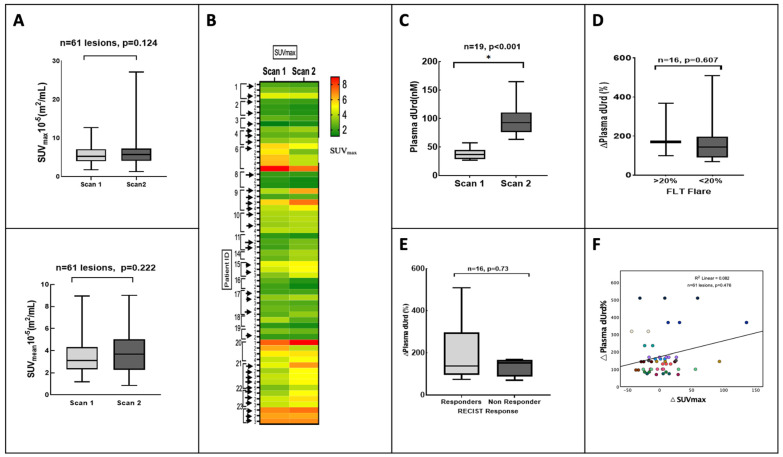
Response heterogeneity detected by ^18^F-FLT uptake and plasma deoxyuridine (dUrd). (**A**) The difference in SUV_mean_ and SUV_max_ between the two scans for all lesions represented as a box plot. (**B**) Visual depiction of the heterogeneity in ^18^F-FLT uptake between the first and second scan in the individual lesions in each patient as seen on a heatmap (lesions with ^18^F-FLT flare shown with black arrow). (**C**) Box plot of the changes in plasma dUrd between the two scans. (**D**) The difference in plasma dUrd among the group with ^18^F-FLT flare above 20% threshold and group with no flare is seen on the box plot (Note * significant *p* value). (**E**) The difference in plasma dUrd among the responders and non-responders based on RECIST 1.1 on imaging depicted by box plot. (**F**) The correlation graph between dUrd change and ^18^F-FLT ∆SUV_max_ for all tumour lesions (as depicted by different colours for each patient with multiple lesions). The Spearman correlation coefficient is R^2^ = 0.082, *p* = 0.476.

**Figure 3 cancers-15-03718-f003:**
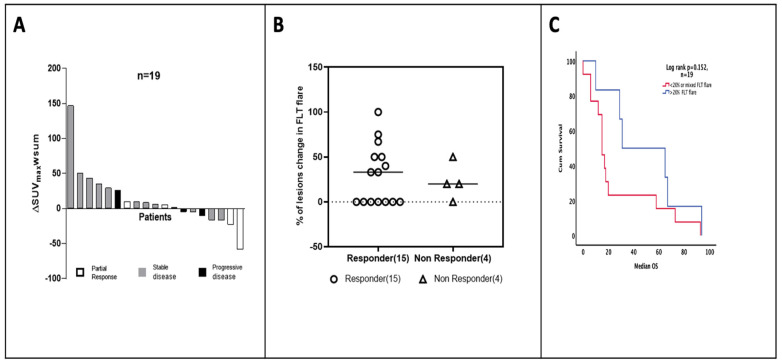
Patient-level comparison of RECIST and clinical outcome with ^18^F-FLT flare. (**A**) The relative tracer uptake represented as waterfall plot of the percentage variation in ^18^F-FLT (∆SUV_max_-wsum) in all patients and their radiological RECIST 1.1 response. (**B**) The percentage variation in the number of lesions per patient with or without a flare among responders and non-responders as shown in the scatter diagram. Responders: progressive disease on follow-up scan. (**C**) Kaplan–Meier survival curve of OS according to ∆SUV_max_ (15 m vs. 31 m, log rank *p* = 0.15, HR 0.48, 95% CI 10.89–25.11). Cut-offs for Kaplan–Meier curve were determined by those above and below the 20% ^18^F-FLT flare threshold.

**Figure 4 cancers-15-03718-f004:**
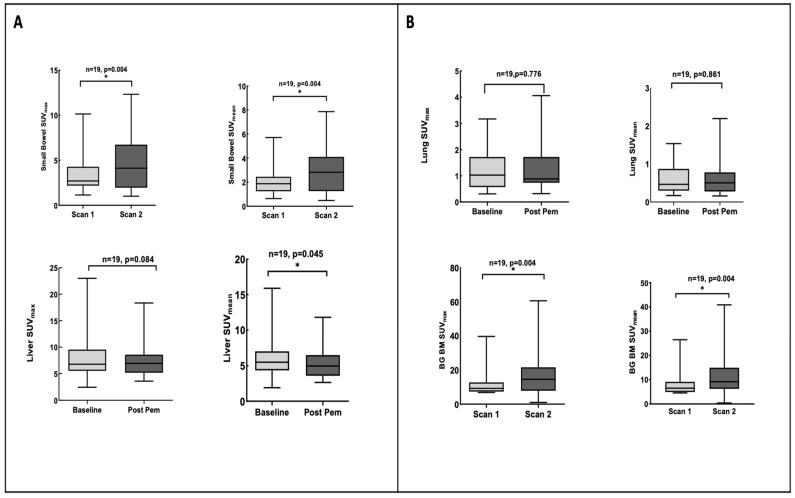
The ^18^F-FLT uptake in background tissues at 4 h post pemetrexed. (**A**) Box plot of the ^18^F-FLT uptake in TS-responsive background tissues, bone marrow, and small bowel at the two scans. Paired *t*-test was significant for small bowel and bone marrow background tissue. (**B**) Box plot of ^18^F-FLT uptake in non-TS-responsive background tissue, lung and liver. * where *p* value is significant.

**Figure 5 cancers-15-03718-f005:**
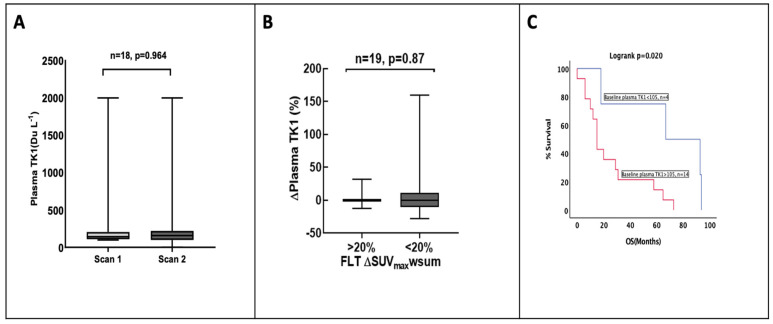
Plasma thymidine kinase 1 (TK1) activity as alternative biomarker to explain changes in ^18^F-FLT. (**A**) The box plot diagram of the change in plasma TK1 activity at the two scan time points. (**B**) Box plot shows the change in plasma TK1 activity among the group with ^18^F FLT flare and no flare. (**C**) Kaplan–Meier survival curve of overall survival (OS) according to baseline plasma TK1 activity (15 m in plasma TK1 activity <105 DuA vs. 67 m in plasma TK1 activity >105 DuA, *p* = 0.022), HR 5.19 (95% CI 7.61–28.39).

**Table 1 cancers-15-03718-t001:** Baseline patient characteristics for all twenty-one patients with NSCLC enrolled in the study.

Demographics	N	(%)
Median age	62	38–78 (IQR)
Gender		
Male	9	(43)
Female	12	(57)
Race		
Caucasian	18	(86)
Asian	2	(9.5)
African	1	(5)
Smoking History		
Non-smoker	5	(24)
Ex-smoker	8	(38)
Smoking at time of diagnosis	8	(38)
Clinical		
ECOG PS		
0	3	(14)
2	3	(14)
1	15	(72)
Histology		
Genetic Alterations		
Nil reported	13	(62)
KRAS codon 61	1	(5)
KRAS G12V	1	(5)
ALK fusions	2	(10)
EGFR ex 19 del, T790M	3	(14)
EGFR L858R	1	(5)
TTF1		
Negative	2	(10)
Positive	19	(1)
Chemotherapy		
Cisplatin + pemetrexed	12	(57)
Carboplatin + pemetrexed	9	(43)
Prior local therapies	3	(14)

Note: data are presented as N(IQR) or N(%). Abbreviation: IQR, interquartile.

## Data Availability

With the exception of human PET-CT images, all data are available in the main text or the Appendix A. Because these image data cannot be fully anonymised for sharing, investigators requiring images should contact the corresponding author to agree contractually on data use.

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
