# Peer review of "A Subset of Non-Small Cell Lung Cancer Patients Treated with Pemetrexed Show 18F-Fluorothymidine “Flare” on Positron Emission Tomography"

_cancers, 2023, doi:10.3390/cancers15143718_

Round 1
Reviewer 1 Report
This study has significant clinical value. In clinical practice, the 18F-FLT flare phenomenon is often found, and the article illustrated the reason and conducted in-depth research. But the following minor error or questions need to be answered.
(1) Please check the format of "18F" which should be 18F.
(2) In upper one of Figure 2A, "n=81" should be "n=81 lesions".
(3) In Figure 2, the sequence number labeling is not tidy, please correct it.
(4) In the 2nd paragraph of page 14, there are repeated "of the" in “The rationale for using the change of the of the weighted”, please correct it.
(5) In PET Analysis Part, why is it 40% of the SUVmax threshold" was chosen, instead of 20% or 30%, please explain it.
(6) In PET Analysis Part, which one is "equation 1", please label it.
(7) Please describe the relationship between dUrd and TS to illustrate the reason to measure plasma dUrd level.
(8) Can you please describe what type of 81 lesions are? For example primary or metastatic?
Reviewer 2 Report
The authors presented an interesting article that provides new insights into the mechanism of action and response to treatment with antitumour agents inhibiting DNA precursor synthesis in patients with advanced lung cancer. The results of this work are a first step towards further, larger studies to improve the selection of patients to be treated with pemetrexed.
However, the number of patients analysed is low and should be specified in the limits of the study.
The figures could be improved: in Figure 1a, the lettering inside the boxes is partially covered and shifted; Figure 2 is of poor quality (the size of the individual boxes could be increased) and the letters do not correspond well to the boxes; Figure 3 does not show the letters for each box; in Figure 4, the dimensions of the boxes are not homogeneous.
Please check the entire manuscript for typographical errors (e.g. in the discussion: "This variable needs to be explored in future studies to evalute the heterogeneity among multip[le lesions"; and in the paragraph Analysis by Plasma dUrd: correct the figure from 2E to 2C).
Reviewer 3 Report
The authors demonstrated the clinical impact of FLT flare in NSCLC patients with pemetrexed treatment. I really appreciate the opportunity to review this manuscript, however, I could not find any impact of FLT flare in NSCLC treatment.
All data show no difference in FLT flare between responder and non-responder; it is well established that FLT uptake increases after 4 hours of Pemetrexed, but it is not at all clear what clinical significance this may have. Furthermore, Figure 5C, which is the only one where there appears to be a difference, shows the relationship between prognosis in the two groups by TK1 activity, and Figures 5a and 5b show that there is no association between TK1 activity and FLT flare, so no clinical impact on FLT flare has been demonstrated. There is none. Furthermore, the data in Figure 5c have not been analyzed for other confounding factors such as background factors in the two groups divided by TK1 activity, making it difficult to find any impact on this significant difference.
Round 2
Reviewer 3 Report
No comments.